# Judging the unseen: The impact of onset controllability in shaping perceptions of defendants with traumatic brain injury

Claire Williams[1,2]*, Inesa Ledovskyte[1]

1 School of Psychology, Faculty of Medicine, Health and Life Science, Swansea University, Swansea, United Kingdom, 2 Elysium Neurological Services, Elysium Healthcare, The Avalon Centre, Swindon, United Kingdom

* claire.williams@swansea.ac.uk

## Abstract

Traumatic brain injury (TBI) has been associated with increased risk of criminality, yet very little is known about how individuals with TBI may intersect with the adjudication phase of the criminal justice system. Therefore, the aim of this study was to conduct the first empirical investigation of how individuals with TBI are perceived within the context of a UK magistrates' court, and how the perceived controllability of the onset of injury may influence perceptions and sentencing-related recommendations. 174 participants (60.35% female, mean age=34.86 years) from a general population sample, reflecting diverse employment and education backgrounds, read a fictional transcript of a magistrate sentencing a defendant for an assault charge. Participants were randomly allocated to a single condition (Onset Controllable, Onset Uncontrollable, or no-TBI control), where the onset controllability of the injury was experimentally manipulated. Participants were asked to make sentence related recommendations and to rate the defendant's level of risk and dangerousness, behavioural tendencies, and the extent to which they felt empathy/sympathy towards them. Additionally, their proximity to, knowledge of, and attitudes towards brain injury were assessed. The perceived onset controllability of the TBI was not found to influence perceptions and sentencing-related recommendations. Instead, participants reported feeling more empathetic towards the defendant and rated their behavioural tendencies more favourably if they were described as having sustained a TBI, irrespective of its onset controllability. This suggests that the presence of TBI might evoke strong empathic responses that counteract the tendency to assign blame based on controllability and may also lead to more favourable behavioural perceptions, but that such evaluations are not strong enough to exert an influence on sentencing related recommendations. Consequently, it is possible that the invisible nature of TBI-related disability, coupled with poor public understanding, may mean that information about a defendant's brain injury is overlooked and/or not taken into full account in sentencing related recommendations.

provided the original author and source are credited.

**Data availability statement:** The data file is available on the OSF: https://osf.io/ydc4p/?view_only=48523d-0569e54d07928271b6669b5c4b.

**Funding:** The author(s) received no specific funding for this work.

**Competing interests:** The authors have declared that no competing interests exist.

## Introduction

Traumatic brain injury (TBI) has been associated with increased risk of crime perpetration and criminality [1–4]. Fazel et al. [5] found that TBI leading to hospitalisation was associated with a threefold increase in the risk of violent crime compared to general population controls. Similarly, in a longitudinal birth cohort study, and after adjusting for pre-birth and childhood confounders, Kennedy et al. [6] found that individuals with TBI were more likely to have committed at least one offence compared to controls. A significantly higher prevalence of TBI has also been reported in both juvenile and adult offender populations [1,7,8]. For example, Hughes et al. [9] reported prevalence rates of brain injury amongst incarcerated youth of between 16.5% and 72.1%, and McMillan et al. [10] found that 78% (85 out of 109) of incarcerated adult women recruited reported a history of significant head injury. To provide context, the lifetime prevalence of TBI in the general population has been reported to fall between 2% and 38.5% [2]. Thus, whilst the reported rates of brain injury vary across studies, evidence consistently suggests that the rate of TBI is proportionally much higher in prison and offender populations.

There is also evidence that incarcerated individuals who have a history of TBI are more likely to incur disciplinary charges [11] and behavioural infractions [12], exhibit reduced ability to maintain rule-abiding behaviour during incarceration [13], and have a higher risk of recidivism compared to those without TBI [14]. Consequently, an important consideration is how individuals with brain injury intersect with the Criminal Justice System (CJS), where the 'hidden' disabilities associated with TBI may result in barriers to accessing justice. Indeed, a lack of awareness and specialist service provision within the CJS could result in a failure to identify and appropriately support individuals with a history of TBI who have offended. For example, they may have trouble engaging with and completing court orders, their behaviour may be misinterpreted or misattributed to other causes, and they may also struggle to comprehend legal processes, communicate effectively with legal personnel, and adhere to probation or parole conditions [15–18].

Accordingly, research has increasingly explored how individuals with TBI intersect with the different stages of the CJS, including corrections, policing, and parole. For instance, assessment tools have been developed to screen for previously undetected brain injury upon entry to prison [19–21], and tailored supports have also been introduced to better support individuals with TBI while in custody [22–24]. For instance, the Brain Injury Linkworker Service [23] works with prisoners who have screened positive for brain injury to assess their needs and develop a comprehensive support plan. In addition to providing education on brain injury to support better integration while in custody and enhance outcomes of offender rehabilitation, a key component is liaison with families, prison staff and external agencies. This liaison provides guidance on how best to support those with brain injury on release, facilitating discharge processes and successful community-reintegration.

In contrast, there is dearth of research exploring the adjudication phase of the CJS, including how individuals with brain injury are perceived and evaluated by legal decision makers (e.g., magistrates, judges, jurors), and how brain injury and related

disability is factored into key legal decisions and related recommendations (e.g., verdict, mitigation, culpability, sentencing, legal responsibility). This is surprising as even though research has consistently shown how lay persons and non-expert professionals hold significant misconceptions about the long term consequences of TBI [25–28], legal decision makers are routinely asked to interpret and synthesise complex information and medical evidence when reaching a verdict and adjudicating on issues such as responsibility, which could encompass information about brain injury and its resultant sequalae. Further, research has shown how nonevidential and/or extra-legal information can significantly influence jurors' decisions, including attitudes and perceptions of individual characteristics such as race and gender, as well information about medical and/or mental health diagnoses [29–32]. Thus, an important consideration is how individuals with brain injury are perceived and evaluated within the CJS, and how information about brain injury may impact legal adjudication.

Gurley & Marcus [33] presented mock jurors with a fictional case summary and psychological testimony in which a defendant diagnosed with either psychosis or psychopathy, was pleading not guilty by reason of insanity (NGRI). Mock jurors were more likely to render a verdict of NGRI if the defendant was diagnosed with psychosis, but also when the onset of the defendant's diagnosis was attributed to a TBI sustained in a road traffic accident six months prior to the crime, compared to when the onset was described as having occurred during adolescence. However, as TBI was not presented or considered independently of the defendant's psychiatric diagnosis, the isolated effects of the defendants TBI on the juror's verdicts cannot be determined.

In contrast, Pierre & Parente [34] investigated how mock jurors evaluate defendants without TBI compared to those with mild or severe TBI in the absence of psychiatric disorder. Irrespective of crime severity (assault versus murder charge), participants perceived the behaviours of the crime committed by the defendant with severe TBI to be higher in morality (i.e., morally acceptable, justifiable, ethical) compared to the defendant with either mild or no TBI. Participants also rated the defendant with severe TBI as less guilty overall compared to the defendant with no TBI. Although, there was also a significant interaction between TBI presentation and crime severity, with the defendant with severe TBI only judged to be significantly less guilty when the charge was murder. Punishment ratings also tended to be less severe overall for the defendants with severe TBI but were only significantly lower for the charge of murder. Finally, recommended punishments also tended to be less severe for the defendants with TBI, with mock jurors typically rating rehabilitation (severe TBI) and community service (mild TBI) as being the most appropriate form of punishments among the TBI conditions. However, as mock jurors were made aware of the study aim prior to taking part and were also assigned to one crime severity but rated all three defendants, Pierre & Parente conducted a second experiment where mock jurors rated a single defendant (Severe, Mild, No TBI) from one crime severity (assault or murder). Defendants who committed the crime of murder were generally perceived to be less moral and more guilty compared to defendants who had committed an assault, and recommended punishments were again found to favour rehabilitation over incarceration for the defendant with severe TBI. However, guilt ratings were not found to significantly differ across the three defendants, and morality ratings were also highest for the defendant with mild TBI. Thus, findings were not wholly consistent across the two experiments, but nevertheless raise the possibility that the presence of TBI may serve as a mitigating factor affecting broader and verdict related perceptions.

Even so, the nascent literature on this topic has only considered the presence or absence of TBI and has not considered how extra-legal and other factors (e.g., familiarity with, attitudes towards, and understanding of brain injury) may influence perceptions. Additionally, it has largely been positioned in the context of decisions concerning responsibility and capital punishment, which are neither representative of most criminal cases, nor judicial decision making outside of the Crown Court system and/or American judicial system. Indeed, such mock juror research can only focus on a small proportion of crimes, and notably the most serious offences committed by offenders, as only 5–10% of crimes are trials by jury. In the United Kingdom (UK), for example, most criminal cases are not only adjudicated in a magistrate's court, but 95% are also completed there [35]. Thus, research exploring the potential role of brain injury in legal decision making that reflects more typical judicial processes in the UK and elsewhere, are sorely needed.

Additionally, no empirical investigations have explored whether the way in which information is presented about TBI, or details about the context of the injury that might be irrelevant to the case at hand, could influence perceptions. One potentially influential piece of information pertains to the perceived controllability of the onset of injury. Drawing on attributional theories of stigma, onset controllability refers to *"the capacity to volitionally alter a cause"* [36] and is closely related to the concept of responsibility. That is, onset controllable causes are those which someone can change or are perceived to be attributable to a lack of will or effort, whereas uncontrollable causes are not subject to personal decisions, behaviour, or management. For example, the onset of heart disease is construed as more onset controllable if the person has led an unhealthy lifestyle, including smoking, a poor diet and lack of physical activity, whereas it is construed as comparatively onset uncontrollable if hereditary factors played a major role in the illness. In turn, construing an illness or condition as controllable or preventable, allows individuals to proportion blame and responsibility to sufferers, as they are deemed to have greater causal responsibility [37,38].

In line with this, several studies have experimentally manipulated onset controllability to determine its impact on perceptions of, and behavioural intentions towards, individuals with various conditions. Tyrrell et al. [39] found that participants rated an individual with spinal cord injury as being warmer (e.g., friendly, well-intentioned, trustworthy, sincere) when their injury was described as onset uncontrollable (struck by car driven by an intoxicated driver) than onset controllable (car accident whilst driving intoxicated). Notably, perceived warmth has been shown to determine whether someone is deemed to have positive intentions and prosocial traits [40]. Consistent with The Belief in a Just World theory [41], which posits that people need to believe that the world is just and fair, and that people get what they deserve, research has also shown that people are less empathetic, sympathetic and willing to help an individual suffering from a condition that is construed as onset controllable [38,42,43]. Evidence also indicates that people are more likely to support punitive policy measures related to conditions perceived as onset controllable [44].

However, few studies have examined onset controllability in the context legal decision-making. Doyon [45] examined responses to an insanity verdict, where participants read a short case description in which a defendant had either been found NGRI or guilty despite entering an NGRI plea. The basis of the NGRI plea was either the presence of an onset controllable (alcoholism or depression) or uncontrollable (Schizophrenia or Post-Traumatic Stress Disorder) condition. Participants assigned greater responsibility to defendants presented with onset controllable than uncontrollable conditions, but only for the crime of homicide versus arson. Moreover, participants were more likely to rate a guilty than NGRI verdict as most appropriate, and experience less pity and more anger, in response to the onset controllable than uncontrollable conditions.

Further, Heath et al. [46] found that simulated jurors were more likely to recommend a guilty verdict and marginally longer sentences when a defendant offered an excuse high (cocaine dependency) rather than moderate (parental abuse) or low (Post-Traumatic Stress Disorder) in onset controllability. Additionally, the excuse offered in the high onset controllability condition was rated as significantly less credible and persuasive. Similarly, Higgins et al. [47] found that participants were more likely to render a guilty verdict and recommended longer sentences when a defendant offered a high (cocaine dependency) versus low (Post-Traumatic Stress Disorder) self-inflicted excuse for their crime.

Taken together, such findings suggest that onset controllability may influence legal related perceptions and recommendations, but whether such effects extend to other conditions, including brain injury, is unclear. It is possible that a TBI presented as onset controllable compared to onset uncontrollable may negatively affect perceptions and evaluations, with legal-decision makers rendering harsher verdicts as a result. Further, there are significant methodological shortcomings attached to prior research examining onset controllability and excuse defences. Rather than directly manipulating onset controllability, existing studies [45–47] have compared different disorders (e.g., cocaine dependency, Post-Traumatic Stress Disorder) that vary in their average levels of onset controllability. Moreover, such conditions invariably differ in terms of their severity and perceived credibility, and public perceptions, attitudes and understanding of each have also been shown to differ. Angermeyer & Matschinger [48] found that people with schizophrenia are far more likely to be considered as dangerous and unpredictable, whereas depression evokes more pro-social reactions in comparison. Kilian et

al. [49] also found that compared to substance-unrelated disorders, people with alcohol use disorders were less likely to be considered mentally ill and were perceived as more dangerous, with other research [50] indicating how perceptions of risk and dangerousness partially mediate the relationship between liability judgements and both recidivism and crime severity. Thus, the manipulation of onset controllability in prior research has been indirect and potentially confounded. Instead, to isolate the effects of onset controllability more precisely, the presented excuse (i.e., the condition, such as TBI) should be held constant across experimental conditions.

In sum, despite the high prevalence of brain injury among individuals in the CJS, very little empirical research has explored how individuals with brain injury intersect with the adjudication phase of the CJS. Further, no existing research has examined the impact of onset controllability of brain injury or the specific context of the magistrates' court in this intersection. Therefore, the overarching aim of the current study was to address this significant gap in understanding by exploring, for the first time, how individuals with TBI are perceived within the context of a UK magistrates' court, and how contextual information (i.e., onset controllability) about a defendant's TBI may influence perceptions, including sentencing-related recommendations. It was hypothesised *a priori* that severity of sentence ratings, sentencing recommendation, empathy/ sympathy towards the defendant, perceptions of risk and dangerousness, and perceptions of the defendant's behavioural tendencies would vary as a function of experimental condition (Onset Controllable, Onset Uncontrollable, No-TBI Control).

Specifically, as prior research suggests that onset controllability may serve as an aggravating factor resulting in reduced empathy [37,38], more negative behavioural evaluations and punitive recommendations [46,47], but that the presence of a TBI [34] may equally serve as a mitigating factor in verdict related decisions and also elicit more empathy when uncontrollable in nature, it was predicted that: (1) harsher and more severe sentence related recommendations would be assigned to the defendant in the Onset Controllable compared to the no-TBI control condition, and in the no-TBI control compared to Onset Uncontrollable condition; (2) the defendant would be rated as significantly more dangerous in the Onset Controllable than Onset Uncontrollable condition, and in the Onset Uncontrollable than no-TBI Control condition; (3) participants would feel more empathetic/sympathetic towards the defendant in the Onset Uncontrollable versus no-TBI control condition, and towards the defendant in the no-TBI control than Onset Controllable condition, and (4) the defendants behavioural tendencies would be viewed more negatively in the Onset Controllable than Onset Uncontrollable condition, and in the Onset Uncontrollable than no-TBI Control condition.

## Materials and methods

### Participants

Participants were drawn from the general adult population between 02/06/2022 and 26/04/2024 and were recruited from Prolific (www.Prolific.co), social media, and via a university subject pool system. Participants had to be at least 18 years of age, resident in the UK, and report English as their first language. Participants recruited via Prolific also had to have a minimum approval rating of 90% (i.e., at least 90% of their submissions to previous studies had been approved) and must not have taken part in any related research by the researchers. Prolific participants received a payment of £2.67, and subject pool participants received two credits in return for participation. G-Power calculations indicated that for 80% power, with a rejection criterion of $p < .05$ and a medium effect size ($f = 0.25$), that a minimum sample size of 159 (53 per condition) was required.

Of the 248 participants who accessed the study, 74 were excluded *a priori* for the following reasons: not meeting the advertised eligibility criteria ($n = 7$), accessing the study but not engaging further ($n = 14$), not providing consent ($n = 1$), not completing the study in full ($n = 36$), failing one or more basic (e.g., selection option 'disagree') attentional check questions ($n = 10$), and failing two or more scenario-based attentional check questions ($n = 6$). Thus, the final sample consisted of 174 participants, of whom 105 (60.35%) identified as female. Ages ranged from 18 to 74 years ($M = 34.86$; $SD = 13.64$), with further demographic information presented in Table 1. Options for Gender and Ethnicity were based on the categories

**Table 1. Summary of demographic characteristics.**

| Demographic Characteristics | Sample (*n*=174) | |
|---|---|---|
| | *n* | *%* |
| Gender | | |
| Male | 66 | 37.93 |
| Female | 105 | 60.35 |
| Non-Binary/ Third Gender | 2 | 1.15 |
| Prefer Not to Say | 1 | 0.58 |
| Ethnicity | | |
| White/Caucasian | 159 | 91.38 |
| Asian (East and South) | 5 | 2.87 |
| Middle Eastern | 2 | 1.15 |
| Black/African American/Caribbean/Black British | 3 | 1.72 |
| Mixed/Multiple Ethnic groups | 1 | 0.58 |
| Prefer Not to Say | 4 | 2.30 |
| Education | | |
| No Formal Education | 1 | 0.58 |
| Secondary/High School | 17 | 9.77 |
| College/Sixth Form | 50 | 28.74 |
| University | 73 | 41.95 |
| Postgraduate | 29 | 16.67 |
| PhD/Doctorate | 4 | 2.30 |
| Employment Status | | |
| Full Time | 68 | 39.10 |
| Part Time | 40 | 22.99 |
| Not Working Due to Disability/Health Reasons | 6 | 3.45 |
| Unemployed | 10 | 5.75 |
| Retired | 5 | 2.87 |
| Unpaid Family Work/Carer | 9 | 5.17 |
| Full Time Student | 34 | 19.54 |
| Part Time Student | 1 | 0.58 |
| Volunteer | 1 | 0.58 |

recommended by the UK Government, with gender referring to the socially constructed roles, behaviours, and identities of female, male and gender-diverse people [51].

## Design

A single-factor, between-subjects design where the independent variable was onset controllability: Onset Controllable (*n*=56), Onset Uncontrollable (*n*=61), and No-TBI Control (*n*=57). The central dependent variables were Severity of Sentence, Sentence Recommendation, Perceived Dangerousness, Empathy/Sympathy, and Behavioural Perceptions of the defendant. Three measures were also included to assess levels of proximity to, knowledge of, and attitudes towards brain injury.

## Materials

Participants supplied demographic details and were asked to reconfirm their eligibility to take part. Participants were then informed that they were about to read a fictional transcript of a magistrate sentencing a defendant in a court of law in

the United Kingdom. They were asked to ensure they were somewhere quiet and free of distractions and informed that each page was set to a timer, meaning that they could not advance to the next page until a minimum amount of time had passed. Timers were set at approximately half of the estimated reading time [52].

**Fictional magistrates' scenario.** Participants were presented with fictional sentencing remarks, ostensibly from a magistrates' court. An example case from the sentencing council [53] served as a template, and input was also obtained from a serving Magistrate to ensure that the materials accurately reflected the deliberation process that takes place within a magistrates' court. Care was taken to avoid providing any specific motivations for the given crime (e.g., a prior relationship between the defendant and victim), and no reference was made to extra-legal variables (e.g., race, socioeconomic status) shown to influence decision making. The magistrate's scenario was presented as follows:

**Magistrates summary.** The scenario described a 22-year-old male defendant who had pleaded guilty to assaulting a 23-year-old male victim, resulting in the charge of Assault Occasioning Actual Bodily Harm contrary to section 47 of Offences against the Person Act 1861 [54] and Crime and Disorder Act 1998 [55]. This type of offence was chosen because it was appropriate for a magistrates' court but also represented the type of crime (violent crime and assaults) most committed by those with TBI [14]. The scenario outlined how an assault had occurred outside of a theatre after a musical performance on the night of 4th September 2021. The incident involved a verbal argument between the defendant and victim, culminating in the defendant physically assaulting the victim.

**Aggravating and mitigating factors.** Reflecting those typically considered by a magistrate when determining a sentence, a set of core aggravating and mitigating factors were presented. These were common across all three experimental conditions and were presented in a counterbalanced order (i.e., aggravating, or mitigating block first). The three aggravating factors were: (1) the defendant repeatedly assaulted the victim (i.e., it was a sustained attack and did not stop after a single blow); (2) it was an unprovoked attack, and (3) the defendant caused significant harm to the victim. The three mitigating factors were: (1) it was the defendant's first offence; (2) the defendant pleaded guilty at the earliest available opportunity, and (3) the offence was not premeditated.

Participants in the Onset Controllable and Onset Uncontrollable conditions were also presented with an additional mitigating item tailored to their specific condition. Care was taken to ensure that the descriptions were comparable in length, style, and overall presentation. First, information pertaining to the circumstances in which the TBI was acquired were presented, including the critical manipulation of onset controllability. The defendants TBI was described as having occurred during a work-related incident in both conditions (e.g., '*You suffered a traumatic brain injury during a work-related incident 14 months ago where an object fell onto your head from above')*, but allocation of responsibility differed.

In the Onset Controllable condition, the critical sentences read: "*At the time of the incident, you were not wearing the full personal protective equipment supplied to you by your employer, even though you were required to wear this as part of your role and as part of your workplace Health and Safety Regulations. A subsequent investigation determined that the injuries you sustained could have been avoided if you had chosen to wear your full personal protective equipment*".

In the Onset Uncontrollable condition, the critical sentences read: "*At the time of the incident, you were wearing the full personal protective equipment supplied to you by your employer, and as required as part of your role – and as per your workplace Health and Safety Regulations. A subsequent investigation determined that your actions did not contribute towards the injuries you sustained*".

Second, identical information regarding the behavioural, cognitive, and emotional effects experienced by the defendant since their brain injury was briefly outlined: "*As a result of your traumatic brain injury, we understand that you have difficulty reading social situations, have a tendency to act impulsively and make rash decisions, experience difficulty concentrating on tasks and managing competing demands, and also struggle to manage and control your emotions*". Additionally, it was acknowledged that the TBI may have contributed to the defendant's actions on the day of the assault. The description of the defendant's brain injury was validated by a Consultant Clinical Neuropsychologist who has clinical and medico-legal expertise, as containing information which would be relevant in a criminal trial.

 

**Sentencing options available to the magistrate.** The aggravating and mitigating factors present for the case meet the criteria for a category 2 offence (greater harm and lower culpability), meaning that a range of options are available to a magistrate, ranging from a low-level community order to a 51-week custodial sentence. The scenario outlined how magistrates begin at a standard starting point according to the sentencing guidelines set by the Sentencing Council in the UK, and then adjust the sentence based on the aggravating and mitigating factors specific to the case. Participants were informed that for the current offence, magistrates could impose the following sentences: (1) A low-level community order (e.g., carrying out unpaid community service which ranges from 40–300 hours); (2) A suspended custodial (prison) sentence of up to 26 weeks (the sentence will only be served if the defendant violates the conditions of the suspension during the suspended period which ranges from 6–24 months); (3) A custodial (prison) sentence of up to 26 weeks, and (4) Referral to the Crown Court for consideration of a custodial sentence beyond 26 weeks (up to a maximum of 51 weeks).

**Dependent measures.** Following presentation of the magistrate's scenario, participants completed the following five measures that were developed for this study:

**Severity of sentence.** Participants rated how lenient or severe they thought the sentence should be on a seven-point scale (1 = Least Severe Sentence to 7 = Most Severe Sentence).

**Sentence recommendation.** This measured the sentencing option that the participant considered most appropriate for the defendant. The four sentencing options available to the magistrate were outlined (see section '*Sentencing Options Available to the Magistrate*'), and after selection, a follow-up question was asked to allow participants to indicate their preferred length/severity of sentence by selecting one of a possible 13 choices worded appropriately to the sentencing option selected. These choices were presented in: (1) increments of 20 hours for a low-level community Order (between 40–300 hours); (2) increments of two weeks for a suspended sentence (up to 26 weeks); (3) increments of two weeks for a custodial prison sentence (up to 26 weeks), and (4) increments of two weeks for referral to Crown Court (between 26–51 weeks). Thus, this resulted in 52 categories of increasing severity. For data analysis, this information was coded as one continuous variable which could range from 1 to 52.

**Perceived dangerousness.** Participants rated their level of agreement (1 = Strongly Disagree to 7 = Strongly Agree) to five statements designed to measure the perceived risk and dangerousness of the defendant (e.g., '*The defendant should be considered a danger to society*' and '*The defendant is likely to re-offend*'). Three of the five items were drawn from prior research [50]. A total composite score (7–35) was obtained by summing responses to individual items, with higher scores indicating greater perceived levels of risk and dangerousness. The scale was found to have good internal reliability (Cronbach's α = .82).

**Empathy/sympathy.** Consisting of eight items rated on a seven-point scale (1 = Strongly Disagree to 7 = Strongly Agree), this scale was designed to assess the extent to which participants felt empathy and sympathy towards the defendant (e.g., '*I can easily relate to the defendant's feelings*' and '*I feel sympathetic towards the defendant and their circumstances*'). Individual items were summed to provide a composite total score (8–56), with higher scores indicating higher levels of empathy/sympathy towards the defendant. The internal reliability of the scale was excellent (Cronbach's α = .90).

**Behavioural perceptions.** Consisting of six items rated on a seven-point scale (1 = Strongly Disagree to 7 = Strongly Agree), this scale was designed to assess participants' perceptions of the defendant's broader behavioural tendencies (e.g., impulsivity, recklessness, responsibility). For example, '*The defendant exhibits good self-control*' and '*The defendant is the type of person who acts without first considering the impact of their actions on others*'. After reverse scoring positively worded items, individual items were summed to provide a composite total score (6–42), with higher scores indicating more negative (i.e., less favourable) behavioural perceptions of the defendant. The internal reliability of the scale was good (Cronbach's α = .84).

**Other measures and questions.** In addition to the dependent measures above, participants completed:

**Level of Contact Report (LOC).** The LOC was originally developed to measure familiarity with, and exposure to, severe mental illness [56]. In this study, an adapted version of the 12-item scale was used, wherein the term 'mental illness' was changed to 'brain injury' throughout. Additionally, the words 'or internet' were added to one statement to account for potential exposure through online resources. Participants select all statements that apply to them, with their score (1–12) corresponding to their highest ranked score. For example, if a participant selects '*I have watched a documentary on the television or internet about brain injury*' (Rank = 4) and '*My job involves providing services/treatments for people with brain injury*' (Rank = 8), a final score of 8 is assigned. In addition to ranked scores, responses were also grouped into three categories to provide an index of overall level of proximity to brain injury: (1) low proximity (little or limited proximity to brain injury – ranks 1–4); (2) medium proximity (providing services and work-related proximity – ranks 5–8), and (3) high proximity (personal and familial proximity with individuals with brain injury – ranks 9–12).

**Perception of Brain Injury Measure (PBIM) [57].** Based on a biopsychosocial framework, the PBIM was designed to measure knowledge and understanding of brain injury. It consists of 36 items rated on a seven-point scale (1 = Definitely False to 7 = Definitely True), with a mid-point (4 = Uncertain) denoting uncertainty. 22 items are 'False' (e.g., '*The primary goal of brain injury rehabilitation is to improve physical functioning*') and 14 items are 'True' (e.g., '*The impact of childhood brain injury can take years to become apparent*'). After reverse scoring 'False' items, scores falling below the mid-point of the scale indicate inaccurate knowledge and can be considered a misconception. Individual items are summed to provide a total score (36–252), with higher scores indicating more accurate knowledge and understanding of brain injury. Additionally, the PBIM includes six subscales corresponding to different domains of knowledge and understanding (e.g., Symptoms of Brain Injury; Physical and Medical Expectations, Severity and Impact of Brain Injury), but these were not utilised here. During its development, PBIM total scores demonstrated moderate test-re-test reliability (ICC = .74) and promising results were found for the measure's convergent and divergent validity. Moreover, PBIM total scores have been shown to be sensitive to detecting knowledge differences between professionals working in the field of brain injury and members of the general population. In the current study, the PBIM was also found to have good internal reliability (Cronbach's $\alpha$ = .85).

**Brain Injury Awareness Scale (BIAS) [57].** Designed to measure attitude-based perceptions of brain injury, the BIAS consists of 20 items rated on a seven-point scale (1 = Strongly Disagree to 7 = Strongly Agree). It contains an equal number of negatively (e.g., '*I would feel unsure about mixing and interacting with someone with a brain injury*') and positively (e.g., '*People with brain injury pose no risk to other people*') worded items, with negatively worded items reverse scored prior to analysis. Individual items are summed to provide a composite total score (20–140), with higher scores indicative of more positive attitudes towards brain injury. Four sub-scale scores can also be calculated (e.g., Social Inclusion, Empathy), but these were not utilised here. BIAS total scores have demonstrated moderate test-re-test reliability (ICC = .74), and promising results have been reported for the scale's convergent and divergent validity. In the current study, the BIAS was found to have good levels of internal reliability (Cronbach's $\alpha$ = .82).

**Scenario-based attentional control check questions.** Three scenario-based questions were included to verify participants attention and understanding of material. For each question, participants were required to select one of four multiple-choice answers, with one being correct and the remaining three false (See Table 2). Participants who failed two or more of these checks were excluded ($n$ = 6) from the analysis *a priori*. Of those taken forward to analysis, 27.59% (n = 48) answered two questions correctly, while 72.41% (n = 126) answered all three questions correctly.

**Basic attentional control checks.** Five basic checks (e.g., '*Select option: Strongly Agree*', and '*Select option: Definitely False*') were embedded, with three in the PBIM and two in the BIAS. Participants who failed one or more of these checks were excluded ($n$ = 10) from the analysis.

**Manipulation check.** To evaluate the effectiveness of the experimental manipulation of onset controllability, participants rated their level of agreement (1 = Strongly Disagree to 7 = Strongly Agree) to the following statement: *"The defendant is to blame and is responsible for their own traumatic brain injury and injuries sustained"*. A significant portion

**Table 2. Response accuracy for scenario-based attentional control check questions.**

| Scenario-Based Attentional Control Check Question | *n* | % |
|---|---|---|
| What age was the defendant? | | |
| Below 20 years | 1 | 0.58 |
| **20–29 years** | 169 | 97.70 |
| 30–39 years | 4 | 2.30 |
| 40–49 years | 0 | 0 |
| Where did the defendant hurt the victim? | | |
| **Face and chest** | 166 | 95.40 |
| Arms and wrist | 0 | 0 |
| Back and shoulder | 1 | 0.58 |
| Stomach and ribs | 7 | 4.00 |
| Where did the assault take place? | | |
| Outside a cinema | 14 | 8.05 |
| **Outside a theatre** | 138 | 79.31 |
| Outside a restaurant | 16 | 9.20 |
| Outside a retail centre | 6 | 3.45 |

*Note.* Correct responses are highlighted in bold.

of participants perceived the defendant's responsibility in a manner consistent with the intended manipulation (e.g., Onset Controllable: 53.7% rated responsibility as 5 or above, Onset Uncontrollable: 83.61% rated responsibility as 3 or below). In line with this, the defendant was rated as significantly more to blame and responsible for their brain injury in the Onset Controllable ($M = 4.43$, $SD = 1.43$) than Onset Uncontrollable ($M = 2.21$, $SD = 1.50$) condition, $t(115) = 8.19$, $p < .001$, $d = 1.52$, 95% CI [1.10, 1.92]. The Bayes Factor also indicated decisive evidence in favour of the alternative hypothesis, $BF_{10} = 1.417 \times 10^{+10}$. Therefore, results indicate that the experimental manipulation of onset controllability was successful.

## Procedure

The research was approved by a University School Research Ethics Committee (Reference 5454) and was conducted in accordance with the 1964 Declaration of Helsinki. It was advertised as '*Exploring how decisions are made in the magistrates' court*' and was administered via Qualtrics [58]. On accessing the study, participants were presented with a detailed information page and were asked to provide written informed consent via an electronic consent page with Yes/No check boxes. Non-consenting participants were redirected to the end of the study.

After consenting, participants reconfirmed their eligibility to participate, supplied demographic information, and were informed that they were about to read a fictional transcript of a magistrate sentencing a defendant in a court of law in the UK. Participants were then randomly allocated (evenly presented elements) to an experimental condition before presentation of the fictional magistrate's scenario. This was presented across different pages (with timers) to reduce the amount of text on the screen at any one time and to increase the salience of critical information: (1) Magistrates Summary; (2) Aggravating and Mitigating Circumstances (counterbalanced order), (3) and Sentencing Options Available to the Magistrate. The dependent measures were then presented in fixed order: (1) Severity of Sentence; (2) Sentence Recommendation; (3) Perceived Dangerousness; (4) Empathy/Sympathy, and (5) Behavioural Perceptions. Within the latter three, items were displayed in randomised order.

Participants then completed the Scenario-Based Attentional Control Check Questions in a fixed order, and those allocated to the Onset Controllable or Uncontrollable conditions also completed the manipulation check. All participants then completed the LOC, PBIM and BIAS which were presented in a randomised order. Items within the PBIM and BIAS

– including the basic attentional control checks – were also randomised, whereas items were presented in a fixed order for the LOC. Finally, participants were presented with a debrief page where the aim of the study was outlined further. All measures required a response, and average completion time was 21 minutes and seven seconds.

## Statistical analysis

Statistical analyses were carried out using JASP version 0.18.3 [59] and SPSS version 29.0.2.0 [60]. Parametric tests were performed throughout, with assumption checks conducted prior to statistical analysis. Any violations are reported in the results, and unless otherwise stated, were not violated. Skewness and Kurtosis values were calculated, and values between −2 to + 2 were deemed to indicate normal distribution [61]. To test the assumption of homogeneity of variance, Levene's test for equality of variance was used, with any violations ($p < .05$) to this assumption reported with accompanying remedial actions.

To examine the presence of possible confounding variables, a chi-squared test of independence, Pearson's correlations, and one-way between subjects of analysis of variance's (ANOVA) were conducted. A series of one way between subject ANOVAs were also conducted to explore the effect onset controllability on the dependent variables, with planned pairwise contrasts performed. Where applicable, alpha values were Bonferroni adjusted to control for the family error rate when a family of tests were required (e.g.,.05 divided by number of contrasts). A hierarchical regression analysis was also performed to explore predictors of sentencing related decisions.

Moreover, as null hypothesis significance tests and their accompanying *p*-values have shortcoming attached, Bayes Factors are also reported for the confirmatory analysis. Bayes Factors can be expressed as the degree to which the data supports the alternative over the null hypothesis ($BF_{10}$) or the degree to which the data supports the null over the alternative hypothesis ($BF_{01}$) [62]. $BF_{10}$ is reported for all analyses, and to aid interpretation, $BF_{01}$ is also reported where data supports the null hypothesis. $BF_{10}$ values are interpreted based on the guidelines proposed by Jeffreys [63], whereby values >1 indicate increasing evidence for the alternative over the null hypothesis, and values <1 the reverse.

## Results

### Possible confounding variables

Prior to examining the main effect of onset controllability on the dependent variables, the presence of possible confounding variables was considered. Proximity to (LOC), knowledge of (PBIM), and attitudes (BIAS) towards brain injury were explored to determine if they might influence the outcomes of interest.

LOC scores are ranked from low (1) to high proximity (12), and the modal rank was 4 ('*I have watched a documentary on the television and internet about brain injury*'). LOC scores were then categorised into low, medium, and high proximity categories, with the 'low proximity' category being the modal rank for all experimental conditions (see Table 3). A chi-squared test of independence confirmed that there was no association between experimental condition and LOC Categories, $X^2(4) = 2.18$, $p = .70$, $BF_{10} = 0.021$, $BF_{01} = 47.32$, with very strong support for the null hypothesis of no association.

Additionally, neither PBIM total, $F(2,171) = 0.42$, $p = .66$, $\eta_p^2 = .01$, $BF_{10} = 0.08$, $BF_{01} = 11.95$, nor BIAS total scores, $F(2,171) = 1.61$, $p = .24$, $\eta_p^2 = .02$, $BF_{10} = 0.23$, $BF_{01} = 4.32$, significantly differed between experimental conditions (see Table 4). Furthermore, the relationship between each of the dependent variables and both PBIM and BIAS total scores was also examined with a series of Pearson's correlations. No significant ($p > .05$) correlations were found, with *r* values ranging from −.005 to.145. Collectively, these findings suggest that neither LOC, PBIM nor BIAS scores are likely to serve as confounding variables. Consequently, they were not included as covariates in the subsequent analysis that follows.

### The impact of onset controllability

To explore whether Severity of Sentence, Sentencing Recommendation, Empathy/Sympathy Perceived Dangerousness, and Behavioural Perceptions varied as a function of experimental condition (Onset Controllable, Onset Uncontrollable,

**Table 3. Number and percentage of participants across the LOC categories by experimental condition.**

| LOC Category | Onset Controllable (n = 56) n (%) | Onset Uncontrollable (n = 61) n (%) | No-TBI Control (n = 61) n (%) |
|---|---|---|---|
| Low[a] | 34 (60.71) | 35 (57.34) | 33 (57.90) |
| Medium[b] | 7 (12.50) | 6 (9.84) | 10 (17.54) |
| High[c] | 15 (26.79) | 20 (32.79) | 14 (24.56) |

*Note:* Low[a] = ranks 1–4, Medium[b] = ranks 5–8, High[c] = ranks 9–12.

**Table 4. PBIM and BIAS scores by experimental condition.**

| Measure | Onset Controllable (n = 56) | | Onset Uncontrollable (n = 61) | | No-TBI Control (n = 57) | |
|---|---|---|---|---|---|---|
| | M | SD | M | SD | M | SD |
| PBIM | 165.66 | 18.82 | 167.77 | 16.25 | 168.84 | 20.96 |
| BIAS | 97.11 | 11.32 | 99.02 | 13.58 | 101.23 | 11.56 |

No-TBI Control), a series of one way between subject ANOVA's were conducted with planned contrasts. Descriptives for each dependent variable are presented in Table 5.

**Severity of sentence.** To test the central hypothesis that Severity of Sentence ratings would vary as a function of experimental condition, a one-way between subject ANOVA was conducted. No significant main effect was found, $F(2, 171) = 0.03$, $p = .97$, $\eta_p^2 < .01$, and Bayes Factors indicated strong support for the null hypothesis of no effect, $BF_{10} = 0.06$, $BF_{01} = 16.75$. Planned contrasts (one tailed) revealed that Severity of Sentence ratings were not significantly different between the Onset Controllable and No-TBI Control conditions, $t(171) = -0.12$, $p = .45$, $d = 0.22$, 95% CI [−0.41, 0.46], $BF_{10} = 0.20$, $BF_{01} = 4.98$, or between the no-TBI and Onset Uncontrollable conditions, $t(171) = -0.25$, $p = .40$, $d = 0.002$, 95% CI [- 0.48, −0.37], $BF_{10} = 0.20$, $BF_{01} = 4.98$.

**Sentence recommendation.** Consistent with Severity of Sentence outcomes and given the large significant positive correlation between Severity of Sentence and Sentence Recommendation ratings, $r(169) = .63$, $p < .001$, there was no significant main effect of experimental condition on Sentence Recommendation ratings, $F(2, 171) = 1.01$, $p = .37$, $\eta_p^2 < .01$. Bayes Factors favoured the null hypothesis of no effect, $BF_{10} = 0.14$, $BF_{01} = 7.20$. Planned contrasts (one tailed) revealed no significant difference in Sentence Recommendation ratings between the Onset Controllable and No-TBI Control conditions, $t(171) = 1.22$, $p = .11$, $d = 0.22$, 95% CI [−1.64, 6.95], $BF_{10} = 0.20$, $BF_{01} = 4.98$, or between the No-TBI Control and Onset Uncontrollable conditions, $t(171) = 0.01$, $p = .49$, $d < .01$, 95% CI [−4.18, 4.24], $BF_{10} = 0.20$, $BF_{01} = 4.97$. Therefore, there was no evidence that the critical manipulation of onset controllability affected sentencing related recommendations.

**Table 5. Descriptives for each dependent variable by experimental condition.**

| Dependent Variable | Onset Controllable (n = 56) | | Onset Uncontrollable (n = 61) | | No-TBI Control (n = 57) | |
|---|---|---|---|---|---|---|
| | M | SD | M | SD | M | SD |
| Severity of Sentence | 3.46 | 1.08 | 3.49 | 1.16 | 3.44 | 1.27 |
| Sentence Recommendation | 21.34 | 10.77 | 18.66 | 10.57 | 18.68 | 13.24 |
| Perceived Dangerousness | 18.66 | 6.23 | 18.13 | 4.60 | 18.46 | 6.23 |
| Empathy/Sympathy | 29.41 | 7.64 | 30.46 | 8.41 | 24.86 | 10.39 |
| Behavioural Perceptions | 28.02 | 4.41 | 27.62 | 5.65 | 30.28 | 5.70 |

**Perceived dangerousness.** As Levene's test of equality of variance was significant, $F(2,171) = 3.69$, $p = .03$, Welch's $F$ is reported. There was no significant main effect of experimental condition on Perceived Dangerousness ratings, $F(2, 111.12) = 0.21$, $p = .81$, $\eta_p^2 < .01$, and Bayes Factors provided strong evidence in favour of the null hypothesis of no effect, $BF_{10} = 0.07$, $BF_{01} = 14.93$. Planned contrasts (one tailed) revealed that Perceived Dangerousness ratings were not significantly different between the Onset Controllable and Onset Uncontrollable conditions, $t(171) = 0.56$, $p = .29$, $d = 0.12$, 95% CI [−1.32, 2.38], $BF_{10} = 0.24$, $BF_{01} = 4.20$, or between the Onset Uncontrollable and No-TBI Control conditions, $t(171) = 0.35$, $p = .36$, $d = 0.06$, 95% CI [−2.17, 1.52,], $BF_{10} = 0.21$, $BF_{01} = 4.87$. Therefore, there was no evidence that onset controllability significantly affected perceptions of the defendant's risk and dangerousness.

**Empathy/sympathy.** The extent to which participants felt Empathy/Sympathy towards the defendant was explored, and Welch's $F$ test is reported as Levene's test of equality of variance was significant, $F(2, 171) = 3.71$, $p = .03$. A significant main effect of experimental condition on Empathy/Sympathy ratings was found, $F(2, 112.15) = 5.43$, $p = 0.01$, $\eta_p^2 = .07$, although Bayes Factors favoured the null hypothesis of no effect, $BF_{10} = 0.07$, $BF_{01} = 15.20$. Planned contrasts (one tailed, adjusted alpha level of .016) revealed that Empathy/Sympathy ratings were significantly higher in the Onset Uncontrollable than No-TBI condition, $t(171) = -3.42$, $p < .001$, $d = 0.60$, 95% CI [−2.37, 8.83], $BF_{10} = 19.01$, but significantly lower in the No-TBI than Onset Controllable condition, $t(171) = -2.72$, $p = .004$, $d = 0.50$, 95% CI [−7.85, −1.25], $BF_{10} = 4.42$. Empathy/Sympathy ratings did not differ between the Onset Uncontrollable and Onset Controllable conditions, $t(171) = 0.64$, $p = .26$, $d = 0.13$, 95% CI [−2.20, 4.29], $BF_{10} = 0.25$, $BF_{01} = 4.06$. Thus, participants were significantly more empathetic/sympathetic towards the defendant when they were presented as having sustained a TBI, regardless of whether the injury was described as onset controllable or uncontrollable.

**Behavioural perceptions.** Participant's perceptions of the defendant's behavioural tendencies (e.g., impulsivity, recklessness, responsibility) were examined. There was a significant main effect, $F(2, 171) = 4.24$, $p = .02$, $\eta_p^2 < .05$, $BF_{10} = 0.45$, but planned contrasts (one tailed, adjusted alpha level of .016) revealed that Behavioural Perception ratings did not significantly differ between the Onset Controllable and Onset Uncontrollable conditions, $t(171) = 0.40$, $p = .34$, $d = 0.08$, 95% CI [−1.54, 2.33], $BF_{10} = 0.21$, $BF_{01} = 4.70$. However, Behavioural Perception ratings were significantly lower (i.e., more favourable/ less negative) in the Onset Uncontrollable compared to No-TBI Control Condition, $t(171) = -2.72$, $p = .003$, $d = 0.47$, 95% CI [−4.59, −0.73], with substantial evidence in favour of the alternative hypothesis, $BF_{10} = 3.46$. Ratings were also significantly lower in the Onset Controllable versus No-TBI Control condition, $t(171) = -2.27$, $p = .01$, $d = 0.44$, 95% CI [−4.23, −0.30], with the Bayes Factor providing some support for the alternative hypothesis, $BF_{10} = 2.35$. Thus, findings suggest that onset controllability did not significantly affect how the defendant's behavioural tendencies were perceived. Instead, participants rated the defendant's behavioural tendencies less negatively when a TBI was present, and irrespective of whether the TBI was described as onset controllable or onset uncontrollable.

## Exploratory analysis

As an exploratory analysis, the relationship between Behavioural Perceptions and Empathy/Sympathy ratings was considered. A significant moderate negative correlation was found, $r(169) = -.43$, $p < .001$, with the Bayes factor providing decisive evidence for the alternative hypothesis, $BF_{10} = 2.257 \times 10^{+6}$. Thus, participants were less likely to empathise/sympathise with the defendant if they viewed their behaviour more negatively (i.e., less favourably) and vice versa.

Whether sentence related recommendations could be predicted from Perceived Dangerousness, Empathy/Sympathy, and Behavioural Perception ratings, as well as Onset Controllability, was also explored. Hierarchical regression analyses were conducted with Severity of Sentence (Model 1) or Sentence Recommendation (Model 2) ratings as the dependent variable, Perceived Dangerousness, Empathy/Sympathy, and Behavioural Perceptions ratings as predictors in step one, and Onset Controllability in step two (see Table 6). In the first step, the predictors accounted for a significant amount of variance in Severity of Sentence, $F(3,170) = 20.49$, $p < .001$, $R^2 = .27$, $R^2_{adj} = .25$, and Sentence Recommendation ratings, $F(3,170) = 23.42$, $p < .001$, $R^2 = .29$, $R^2_{adj} = .28$. Perceived Dangerousness ratings were a significant unique predictor in both

**Table 6. Hierarchical regression analysis predicting sentence related recommendations.**

| Model and Step | B | SE B | β | t | p |
|---|---|---|---|---|---|
| Model 1: Severity of Sentence | | | | | |
| Step 1 | | | | | |
| Perceived Dangerousness | 0.08 | 0.02 | 0.36 | 4.37 | <.001 |
| Empathy/Sympathy | −0.01 | 0.01 | −0.09 | −1.23 | .22 |
| Behavioural Perceptions | 0.04 | 0.02 | 0.17 | 1.98 | .05 |
| Step 2 | | | | | |
| Perceived Dangerousness | 0.08 | 0.02 | 0.34 | 4.08 | <.001 |
| Empathy/Sympathy | −0.01 | 0.01 | −0.11 | −1.50 | .14 |
| Behavioural Perceptions | 0.04 | 0.02 | 0.19 | 2.18 | .03 |
| Onset Controllable | 0.17 | 0.20 | 0.07 | 0.84 | .40 |
| Onset Uncontrollable | 0.27 | 0.20 | 0.11 | 1.36 | .18 |
| Model 2: Sentence Recommendation | | | | | |
| Step 1 | | | | | |
| Perceived Dangerousness | 0.99 | 0.19 | 0.41 | 5.04 | <.001 |
| Empathy/Sympathy | −0.08 | 0.10 | −0.09 | −0.92 | .36 |
| Behavioural Perceptions | 0.26 | 0.18 | 0.12 | 1.46 | .15 |
| Step 2 | | | | | |
| Perceived Dangerousness | 0.93 | 0.19 | 0.41 | 5.04 | <.001 |
| Empathy/Sympathy | −0.11 | 0.09 | −0.09 | −1.21 | .23 |
| Behavioural Perceptions | 0.31 | 0.18 | 0.15 | 1.75 | .08 |
| Onset Controllable | 3.69 | 1.92 | 0.15 | 1.93 | .06 |
| Onset Uncontrollable | 1.74 | 1.90 | 0.07 | 0.92 | .36 |

models, and Behavioural Perception ratings just reached statistical significance in Model 1. In the second step, the overall models remained significant for both Severity of Sentence, $F(5,168) = 12.6$, p < .001, $R^2 = .27$, $R^2_{adj} = .25$, ΔR² Change = .01, $R^2_{adj} = .25$, and Sentence Recommendation ratings, $F(5,168) = 14.94$, p < .001, $R^2 = .31$, $R^2_{adj} = .29$, ΔR² Change = .02. However, the introduction of Onset Controllability did not account for additional significant variance above and beyond the variables entered in the first step. In the second step, Perceived Dangerousness ratings remained a significant unique predictor in both models, as well as Behavioural Perception ratings in Model 1.

## Discussion

The overarching aim of the study was to explore how individuals with TBI are perceived within the context of a UK magistrates' court, and how contextual information (i.e., onset controllability) about a defendant's TBI may influence perceptions and sentencing-related recommendations. In contrast to prior research [45–47], no significant main effect of onset controllability was found on severity of sentence or sentence recommendation ratings, and the Bayes Factors strongly supported the null hypothesis of no effect. Additionally, onset controllability did not emerge as a significant unique predictor of sentence related recommendations, nor did it account for a significant amount of additional variance in the exploratory regression analysis. These findings suggest that within the context of TBI, the perceived controllability of the injury does not significantly affect participants sentencing related recommendations.

Our contrasting findings may be attributable, at least in part, to the methodological rigour in which the effects of onset controllability were isolated in the current study. As explained previously, there are significant methodological shortcomings attached to prior research examining onset controllability and excuse defences. That is, rather than directly manipulating onset controllability, existing studies [45–47] have compared different disorders that vary in their average levels of

onset controllability, severity, and perceived credibility (see S1 Table for a comparative summary of the key differences and similarities between the current study and past research on onset controllability). Thus, the manipulation of onset controllability in prior research has been indirect and potentially confounded. In contrast, the presented condition here (i.e., TBI) was held constant across the onset controllable and uncontrollable conditions. Therefore, it is possible that onset controllability might not have as pronounced effect on legal related judgements as previously reported when effects are isolated more robustly.

Alternatively, the emotional and moral salience of the current experimental manipulation may offer a complementary explanation. Participants in the current study may have perceived the behaviour leading to injury (e.g., failure to wear protective equipment) as relatively minor and less morally reprehensible, in contrast to the more blameworthy behaviours used in prior research (e.g., cocaine dependency, driving while intoxicated). Consequently, previously reported effects of onset controllability may have been driven by the moral salience and perceived culpability of the antecedent behaviour, rather than by controllability per se. Indeed, whilst our experimental manipulation successfully shifted perceptions of controllability and responsibility, these judgements may not have translated into sentencing or empathy-related decisions. That is, whilst the experimental manipulation may have influenced attributions of responsibility, it may not have been impactful enough to shape more complex, consequential judgements – particularly if participants viewed the injury in both conditions as the result of an unfortunate one-time event, rather than a repeated pattern of risky or morally questionable behaviour.

It may also be that the severity and nature of TBI as a serious medical condition with emotional, behavioural and cognitive impacts on an individual, eclipsed considerations of controllability in the current study, leading participants to focus more on the injury itself rather than its onset. However, this is not a strong explanation as no significant differences in sentencing related recommendations were found between the no-TBI control and conditions involving TBI. Indeed, if the presence of TBI serves as a potential mitigating factor leading to more lenient sentencing recommendations as previously suggested [33,34], then significant differences between the conditions involving TBI and the no-TBI control condition might have been expected. Additionally, previous research has consistently shown how laypersons and non-expert professionals often hold significant misconceptions about the long-term consequences of TBI [26,27], and that prior familiarity and exposure to brain injury is associated with fewer misconceptions [25,28]. Therefore, it is possible that no differences emerged because there was general unfamiliarity and lack of understanding of brain injury amongst participants. In line with this, most participants reported little or limited proximity to brain injury, with the 'low proximity' category of the LOC being the modal rank for all experimental conditions. Therefore, this general lack of familiarity may have hindered participants' ability to fully appreciate the complexity and severity of TBI and subsequently accommodate for such information in their sentencing related judgements.

Furthermore, it is also possible that the severity of the offense may have contributed to the null findings. In the current study, the crime of assault was chosen because it was appropriate for a magistrates' court, where a more serious crime like murder was beyond the scope of what could be included in the magistrates' fictional case materials. However, Pierre & Parente [34] previously found that a defendant with severe TBI was judged to be significantly less guilty compared to a defendant with either mild or no TBI, but only when the charge was murder versus assault. Therefore, it is possible that the severity of crime in the current study did not elicit the same degree of differential judgements that might be observed with a more severe offence. Equally though, Bailis et al. [64] previously found that a higher level of dysfunction was needed to justify a verdict of NGRI when the offence was more rather than less serious. This would suggest that a given excuse and/or circumstance, such as TBI, may be more likely to act as a mitigating factor for less severe crimes. Consequently, how the severity of the crime may affect people's willingness to accept a given excuse or circumstance as a mitigating factor remains unclear. To deepen our understanding of such issues, subsequent research should further explore the interaction between crime severity and both TBI and onset controllability, independently and in combination.

                                    

Additionally, it is also possible that participants may process and weigh mitigating factors differently depending on whether they are determining guilt or considering factors for sentencing. Most prior research on onset controllability and legal decision making [46,47] has asked participants to render a verdict (guilty or not guilty), whereas the defendant had already pleaded guilty in the current study. Thus, participants did not need to consider the information about TBI and its onset controllability in relation to perceived guilt or innocence. Consequently, the impact of the defendant's mitigating circumstances may have been diminished, as the focus was on making severity of sentence related recommendations rather than determining criminal responsibility. Indeed, as the defendant had already pleaded guilty, there was perhaps less room for variation to occur across experimental conditions in how participants perceived the defendant's responsibility for the crime itself. In sum, differences in study design, crime severity, and focus on sentencing related recommendations versus guilt, may partially explain the discrepancy in findings between previous research and the current study.

The impact of onset controllability on perceptions of risk and dangerousness were also examined. On the basis that the presence of TBI and its related sequalae would be perceived as heightening the risk of unpredictable and future negative behaviour, but that such perceptions would be amplified when the injury was onset controllable (i.e., signalling reckless-ness and irresponsibility prior to injury), it was predicted that participants would rate the defendant as significantly more dangerous in the Onset Controllable than Onset Uncontrollable condition, and in the Onset Uncontrollable than no-TBI Control condition. Of course, it is worth noting that as participants may weigh and subsequently balance information about the presence of TBI and its onset controllability across conditions differently, the reverse pattern of perceived risk and dangerousness could arguably have been predicted across the latter two conditions (i.e., no-TBI control > Onset Uncon-trollable). Nonetheless, in both scenarios, the highest levels of perceived risk would be anticipated in the onset controlla-ble condition.

However, there was no evidence that onset controllability within the context of TBI significantly affected perceptions of the defendant's risk and dangerousness. Participants rated the defendant similarly across the three conditions, providing indirect evidence that the presence of TBI, irrespective of its onset controllability, does not lead to heightened perceptions of risk and dangerousness. This is somewhat surprising, as it contrasts with prior literature finding that public understand-ing of TBI is poor, whereby lay persons and non-expect professionals can hold negative and stigmatising views. Linden & Boylan [65] found that members of the public tended towards negative labels (e.g., aggressive, agitated) when asked to describe someone with a brain injury, and Linden & Crothers [66] found that students endorsed negative attributes refer-ring to unpredictability, irresponsibility and aggression to characterise survivors of brain injury. One possibility is that within a legal context, participants may focus more on the specifics on the crime rather than the personal circumstances of the defendant and may not perceive the presence of TBI as a factor elevating future risk without explicit information linking brain injury to increased risk and dangerousness. However, this does not seem like a viable explanation given that jurors have been found to consider future dangerousness even when not asked to explicitly do so [67], and that future danger-ousness has been shown to inform mock juror decisions concerning liability and sentencing [68]. Consistent with this, perceived dangerous ratings were found to be a significant unique predictor of both Severity of Sentence and Sentence Recommendations in the current study, with higher levels of perceived risk associated with increasingly harsher sentenc-ing recommendations.

Further, and in contrast to both theory and prior research [38] suggesting that onset controllable conditions evoke less empathy [38,47], there was also no significant difference in empathy/sympathy ratings between the onset controllable and uncontrollable conditions. Instead, and compared to the no-TBI control condition, participants reported feeling significantly more empathetic towards the defendant if they were described as having sustained a TBI, regardless of whether the injury was described as onset controllable or uncontrollable (see S1 Table). Of course, it is reasonable to expect that individuals will respond more empathetically to an individual who has suffered some form of injury, in much the same way that people generally respond to the distress of others. However, the current findings suggest that when empathy/sympathy is elic-ited, such feelings may overshadow considerations of responsibility and controllability. That is, even if participants failed

to fully appreciate the complexity and severity of the TBI, they may still have viewed the presence of a health condition as significant hardship, leading to increased empathic concern and sympathy regardless of the defendants role in acquiring their TBI (i.e., controllability), and thereby diminishing the impact of attributions of blame and responsibility based on the perceived controllability of the injury.

Similarly, the critical manipulation of onset controllability did not significantly affect perceptions of the defendant's broader behavioural tendencies. The less favourable perceptions of the defendant in the no-TBI condition were expected, as no serious health condition or excuse was present to mitigate against negative behavioural perceptions. However, participants did not differentiate between the onset controllable and onset uncontrollable conditions in their behavioural evaluations. That is, the presence of TBI, irrespective of its onset controllability, led to more favourable perceptions of the defendant's behaviour compared to the no-TBI control condition. In the onset uncontrollable condition, participants may have perceived the defendant's behaviour primarily in context of the TBI itself, thus attributing behaviours to the uncontrollable injury rather than the defendant's inherent behavioural traits. In contrast, whilst participants may have perceived the defendant in the onset controllable condition as perhaps having more control over their behaviour and having greater responsibility for their actions, the empathy elicited by the presence of the TBI may have mitigated against negative behavioural evaluations. Consequently, participants viewed the defendant more positively than they would have without the context of the TBI.

Consistent with this, empathy/sympathy ratings were found to be related to broader behavioural perceptions of the defendant (i.e., recklessness, impulsivity, responsibility), evidenced by the moderate negative correlation between empathy/sympathy and behavioural perception ratings. This finding is generally consistent with Empathy-Altruism type hypotheses, which propose that when people feel more empathetic towards an individual, they are inclined to view their actions less harshly and perceive their behaviour more favourably. However, whilst it is possible that participants were more likely to view the defendant's behaviour more positively if they empathised with their position, it could also be a bidirectional relationship, whereby participants are less likely to empathise/sympathise with the defendant if they viewed their behaviour more negatively.

Moreover, and despite the significant relationship between empathy/sympathy and behavioural perceptions, it is also noteworthy – as discussed previously – that there was no significant main effect of onset controllability on severity of sentence or sentencing related recommendations. Thus, whilst empathy and sympathy towards the defendant may influence broader behavioural perceptions, the current findings suggest that empathy towards the defendant may not subsequently affect perceptions of risk and dangerousness or necessarily translate into less punitive sentencing recommendations. Therefore, the effect of empathy/sympathy may be moderated by other factors in the context of legal decision making. Indeed, although behavioural perception ratings emerged as a significant, albeit weak, predictor of Severity of Sentence outcomes (but not Sentence Recommendation), empathy/sympathy ratings were not found to significantly predict either severity of sentence ratings or sentencing recommendations in the exploratory hierarchical regression analysis. This contrasts previous research reporting that state empathy influences mock juror evaluations of trial information. For example, Wood [69] found that inducing empathy in decision-makers impacted on trial outcomes, with participants with higher state empathy holding a defendant as less responsible for the offence and being more likely to disagree with a guilty verdict. Instead, the current findings suggest that the presence of a serious health condition like TBI, might evoke strong empathic responses that counteract the tendency to assign blame based on controllability and may also lead to more favourable behavioural perceptions, but that such evaluations are not strong enough to exert an influence or serve as a mitigating factor when reaching verdict and sentencing related decisions and recommendations.

Finally, even though there was no significant effect of onset controllability on empathy/sympathy ratings, and although the findings initially appear to contradict the Belief in a Just World theory [41], they can still be interpreted within that theoretical framework. In the Onset Controllable condition for example, participants may have experienced a conflict between feeling empathy and assigning blame. However, they may have also found themselves balancing their emotional response

with their desire to see justice served, potentially moderating the influence of empathy on sentencing related recommendations. In contrast, participants in the Onset Uncontrollable condition may have felt more empathy and sympathy for the defendant but then balanced these feelings with the need for a proportionate legal response still based on the overall circumstances of the case (i.e., perceived culpability and harm caused to the victim). Therefore, even though the presence of TBI elicited empathy irrespective of its onset controllability, it is possible that participants balanced their just world beliefs differently in each condition, where perceptions of justice and need for a proportionate legal response overrode emotional reactions. Future research should explore other potential moderating factors, such as the severity of the offense or individual differences among participants, to better understand the complex interplay between empathy, responsibility, and justice in legal decision-making.

### Limitations and future considerations

First, even though presenting a case scenario is a commonly used method and was also developed to be as plausible and authentic as possible, it may not capture the complexity and emotional intensity of real-life legal decision making. Participants were not exposed to live testimony, expert evidence, non-verbal cues, or the persuasive strategies of legal professionals for example, which could affect perceptions and judgements. Second, and even though representing a notable improvement on existing trial simulation research and similar, which typically relies heavily on student samples (e.g., [34]), members of the general population were sampled here. Whilst magistrates in the UK are peer members of the community who sit in their role in a voluntary capacity [35], systemic differences between members of the general population and magistrates may exist that impact on how they respond to a defendant who presents with a TBI. Therefore, whilst this study offers valuable insight into how individuals with TBI may be perceived during the adjudication phase of the CJS, future research would benefit from trying to sample more closely from the population of interest.

Third, although the manipulation of onset controllability was successful, evidenced by the significant difference in responsibility ratings across the Onset Controllable and Uncontrollable conditions, the wording of the critical sentences in the case scenario could have been more equivalently matched across conditions. In turn, this would mitigate against the risk of introducing subtle differences in how participants interpret blame and responsibility. Fourth, even though proximity to, knowledge of, and attitudes towards brain injury were considered as potential confounding variables, other extra-legal variables (i.e., legal attitudes, trait empathy, previous encounters with the CJS) may have influenced participants' perceptions of the case presented. Future research should aim to account for such variables to further isolate the effects of the critical variables and/or manipulations being explored. Finally, whilst the adopted experimental design was robust and appropriate for examining the effects of onset controllability, including a fourth condition – TBI Control – without reference to any notion of blame or responsibility, would have allowed further insight to be drawn regarding the effects of TBI itself as well as onset controllability. However, creating such a neutral and ambiguous condition is not necessarily straightforward, as participants may still make inferences about blame and responsibility even when not mentioned, or when explicitly instructed not to make such assumptions.

### Conclusions

The present research represents the first empirical investigation of how contextual information (i.e., onset controllability) about a defendant's TBI may influence perceptions and sentencing-related recommendations, providing novel insights about how individuals with brain injury may intersect with the adjudication phase of the CJS. The perceived onset controllability of a TBI was not found to influence perceptions and sentencing-related recommendations, suggesting that onset controllability might not have as pronounced effect on legal related judgements as previously reported when effects are isolated more robustly. Instead, current findings suggest that the presence of TBI might evoke strong empathic responses that counteract the tendency to assign blame based on controllability and may also lead to more favourable behavioural

perceptions, but that such evaluations are not strong enough to exert an influence or serve as a mitigating factor when reaching sentencing related decisions and recommendations.

This has significant implications for legal practice and related policy, where there have been recent initiatives to bring attention to brain injury in sentencing related guidelines. For example, the Sentencing Council in the UK recently added acquired brain injury as a condition in their overarching guidance on 'Sentencing offenders with mental disorders, developmental disorders, or neurological impairments' [70], with their overarching position being that if an offender has an impairment or disorder, it should always be considered by the court and that it may reduce culpability if an offender was at the time of an offence suffering from a condition or disorder that impaired their ability, for example, to exercise appropriate judgement, to make rational choices, and/or to understand the nature and consequences of their actions. However, the current findings suggest that whilst participants noted the presence of TBI (i.e., main effects of empathy, behavioural perceptions), the description of that TBI and its effects on the individual (e.g., impulsivity, making rash decisions), as well as the explicit reference to how it may have contributed to their actions on the day of the assault, did not impact on their sentencing related recommendations. Consequently, it may be that invisible nature of TBI-related disability, coupled with poor public understanding of brain injury, may lead to information about a defendant's brain injury being overlooked and/or not taken into full account in sentencing related recommendations. This underscores the importance of improved education and awareness among legal professionals and the public to ensure that TBI, where appropriate and relevant, is fully considered in legal contexts.

## Supporting information

**S1 Table. Comparative summary and synthesis of findings from onset controllability research in legal contexts.** (DOCX)

## Author contributions

**Conceptualization:** Claire Williams, Inesa Ledovskyte.

**Data curation:** Claire Williams, Inesa Ledovskyte.

**Formal analysis:** Claire Williams.

**Investigation:** Claire Williams, Inesa Ledovskyte.

**Methodology:** Claire Williams, Inesa Ledovskyte.

**Project administration:** Claire Williams.

**Supervision:** Claire Williams.

**Writing – original draft:** Claire Williams.

**Writing – review & editing:** Claire Williams.

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
