## [Editor Report · Decision Letter 0]

29 Jul 2024

Dear Dr. Williams,

Thank you for submitting your manuscript to PLOS ONE. After careful consideration, we feel that it has merit but does not fully meet PLOS ONE’s publication criteria as it currently stands. Therefore, we invite you to submit a revised version of the manuscript that addresses the points raised during the review process.

Adding a clear and concise diagram to justify your findings compared to other researchers will enhance clarity. Highlight key differences and similarities, ensuring the diagram is easily understandable and visually appealing. This will provide a better visual context for your results.

Please submit your revised manuscript by Sep 12 2024 11:59PM. If you will need more time than this to complete your revisions, please reply to this message or contact the journal office at plosone@plos.org . A rebuttal letter that responds to each point raised by the academic editor and reviewer(s). You should upload this letter as a separate file labeled 'Response to Reviewers'.A marked-up copy of your manuscript that highlights changes made to the original version. You should upload this as a separate file labeled 'Revised Manuscript with Track Changes'.An unmarked version of your revised paper without tracked changes. You should upload this as a separate file labeled 'Manuscript'.

We look forward to receiving your revised manuscript.

Kind regards,

Opeyemi Oluwasanmi Adeloye

Academic Editor

PLOS ONE
---

## [Author Response · Author response to Decision Letter 1]

31 Jul 2024

Thank you for your feedback. We have responded to each of your suggestions and queries in the list below, with our responses recorded in blue text/below each point. Please note that page and line numbers in our response correspond to the unmarked version of our manuscript.

(1) Adding a clear and concise diagram to justify your findings compared to other researchers will enhance clarity. Highlight key differences and similarities, ensuring the diagram is easily understandable and visually appealing. This will provide a better visual context for your results.

Thank you for your feedback. In response, we have produced a comparative summary and synthesis table (S1 Table) that summarises key differences and similarities between our study and past research on onset controllability (referenced on lines 621-622 and 707 in the discussion). We have opted for this format instead of a Venn diagram or flow chart, for example, for several reasons. Whilst our study has been informed by prior work on onset controllability, it represents the first time that onset controllability has been explored within the context of TBI. Therefore, the context, design, and dependent measures vary significantly across the present and prior work in the broader area. This variability across studies makes it challenging to create a single, visually appealing diagram that accurately represents the nuances of each without either oversimplifying the information, or risking a complex figure that does not bring additional clarity for readers. Instead, an accessible comparative table has allowed us to present the relevant details more succinctly and clearly, supporting readers to easily compare the different elements of each study (e.g., the manipulations, contexts, sample, methodology) as well as the findings (differences and similarities) across each. As such, it has also allowed us to add further detail to complement both the interpretation of study findings and our corresponding discussion of these findings, and the description and overview of each study cited within the manuscript body.

In addition to the above, we have also taken opportunity to correct a spelling error on line 178 and make a small adjustment to our description of a prior study in the introduction section (line 164).

Journal Requirements

Levels heading and titles (sentence case) have been adjusted accordingly throughout the manuscript body. Additionally, the title page has also been updated (i.e., affiliations by number; formatting of corresponding author details; title in sentence case).

(2) When completing the data availability statement of the submission form, you indicated that you will make your data available on acceptance. We strongly recommend all authors decide on a data sharing plan before acceptance, as the process can be lengthy and hold up publication timelines. Please note that, though access restrictions are acceptable now, your entire data will need to be made freely accessible if your manuscript is accepted for publication. This policy applies to all data except where public deposition would breach compliance with the protocol approved by your research ethics board. If you are unable to adhere to our open data policy, please kindly revise your statement to explain your reasoning and we will seek the editor's input on an exemption. Please be assured that, once you have provided your new statement, the assessment of your exemption will not hold up the peer review process.

The data file is available on the OSF and is already available. Our data availability statement has been updated accordingly to reflect this.

https://osf.io/ydc4p/?view_only=48523d0569e54d07928271b6669b5c4b

The reference list has been reviewed for completeness. In response, two additional references [55, 56] have been added to the reference list (lines 990-994) that were previously omitted. Apologies for this oversight. Additionally, no papers have been cited that have retraction notices.

---

## [Decision Letter · Decision Letter 1]

11 Jul 2025

Dear Dr. Williams,

Thank you for submitting your manuscript to PLOS ONE. After careful consideration, we feel that it has merit but does not fully meet PLOS ONE’s publication criteria as it currently stands. Therefore, we invite you to submit a revised version of the manuscript that addresses the points raised during the review process.

 Could you please revise the manuscript to carefully address the concerns raised?

We look forward to receiving your revised manuscript.

Kind regards,

Katrien G. Janin, PhD

Associate Editor, PLOS One

Journal Requirements:

Reviewers' comments:

Reviewer's Responses to Questions

**Comments to the Author**

Reviewer #1: (No Response)

Reviewer #2: (No Response)

2. Is the manuscript technically sound, and do the data support the conclusions?

Reviewer #1: Yes

Reviewer #2: Yes

3. Has the statistical analysis been performed appropriately and rigorously?

Reviewer #1: Yes

Reviewer #2: Yes

4. Have the authors made all data underlying the findings in their manuscript fully available?

Reviewer #1: Yes

Reviewer #2: Yes

5. Is the manuscript presented in an intelligible fashion and written in standard English?

Reviewer #1: Yes

Reviewer #2: Yes

Reviewer #1: 1. In the results section, the ages (range) was described however was not added to the demographics table, this is a key factor which could influence the course of the outcome.

2. The study, in both the methods or the results makes no mention of the occupation of the participants even though it mentions they are of general population in the supporting information. It's prudent these information are placed in the main write up especially in these two sections if possible and also describe or make mention of it in the abstract. If data was collected on the occupation of the participants they could be categorized and presented in the demographics in the results section.

In all the manuscript by Williams and Ledovskyte presents interesting findings on the perception TBI associated criminality in the UK. Generally the paper is well written and well presented.

Reviewer #2: This is a very relevant, interesting, and methodologically robust piece of research.

My only comment concerns the explanations offered in support of the results of the study under Discussion. The authors propose various possible explanations for the fact that perceived controllability of TBI did not significantly affect participants sentencing related recommendations. However, they do not touch upon the very controllability factor in the hypothetical scenario. Possible explanations could also lie in the perceived controllability and moral dimension of the defendants behaviour prior to the accident, which are much less pronounced in the current study compared to the previous ones. Perhaps the respondents assessed the defendant’s failing to wear a protective equipment prior to the accident as (morally) less reproachable compared to e.g. long-term drug and/or alcohol abuse. Similarly, they might have assessed TBI in both scenarios (onset controllable and onset uncontrollable) as a consequence of an unfortunate one-time coincidence (and thus, in both cases as less controllable), in contrast to a repeating substance abuse behaviour in other studies.

This hypothesis would also explain the study’s further finding that “in contrast to both theory and prior research suggesting that onset controllable conditions evoke less empathy, there was also no significant difference in empathy/sympathy ratings between the onset controllable and uncontrollable conditions”.

Should the authors find this proposition relevant, they might briefly comment on it in discussion.

In lines 258-259 there is a small typing error: “Care was taken was avoid providing any specific motivations...”

**Do you want your identity to be public for this peer review?** For information about this choice, including consent withdrawal, please see our Privacy Policy

Reviewer #1: No

Reviewer #2: **Yes: ** Miha Hafner

---

## [Author Response · Author response to Decision Letter 2]

22 Jul 2025

Thank you for your feedback. We have responded to each of your suggestions and queries in the list below. Please note that page and line numbers in our response correspond to the unmarked version of our manuscript.

Reviewer #1:

(1) In the results section, the ages (range) was described however was not added to the demographics table, this is a key factor which could influence the course of the outcome.

Participant ages ranged from 18 to 74 years (M = 34.86; SD = 13.64), as reported in the participants section of the manuscript (lines 230-231). To avoid unnecessary repetition, and because Table 1 focusses on categorical variables reported as Ns and percentages, we have not added age to demographic characteristics table.

(2) The study, in both the methods or the results makes no mention of the occupation of the participants even though it mentions they are of general population in the supporting information. It's prudent these information are placed in the main write up especially in these two sections if possible and also describe or make mention of it in the abstract. If data was collected on the occupation of the participants they could be categorized and presented in the demographics in the results section.

Thank you for this comment. We did not collect data on participants' specific occupations (e.g., job titles or employment sectors). However, we collected and report a range of demographic characteristics in Table 1 (line 236), including gender, ethnicity, highest educational qualification, and employment status (e.g., full-time, part-time, unemployed, student, retired). These categories provide a broad overview of participants' socio-economic and employment contexts. The range of employment statuses and educational qualifications represents diversity expected from a general population sample. For example, unlike some prior research in this area, our participants were not limited to university students or a single educational group. In line with your suggestion, we have also amended the abstract to clarify that the sample was drawn from the general population (lines 19-20).

(3) In all the manuscript by Williams and Ledovskyte presents interesting findings on the perception TBI associated criminality in the UK. Generally the paper is well written and well presented.

Thank you for your helpful suggestions and positive comments.

Reviewer #2:

(1) This is a very relevant, interesting, and methodologically robust piece of research.

Thank you for your positive comments.

(2) My only comment concerns the explanations offered in support of the results of the study under Discussion. The authors propose various possible explanations for the fact that perceived controllability of TBI did not significantly affect participants sentencing related recommendations. However, they do not touch upon the very controllability factor in the hypothetical scenario. Possible explanations could also lie in the perceived controllability and moral dimension of the defendants behaviour prior to the accident, which are much less pronounced in the current study compared to the previous ones. Perhaps the respondents assessed the defendant’s failing to wear a protective equipment prior to the accident as (morally) less reproachable compared to e.g. long-term drug and/or alcohol abuse. Similarly, they might have assessed TBI in both scenarios (onset controllable and onset uncontrollable) as a consequence of an unfortunate one-time coincidence (and thus, in both cases as less controllable), in contrast to a repeating substance abuse behaviour in other studies.

This hypothesis would also explain the study’s further finding that “in contrast to both theory and prior research suggesting that onset controllable conditions evoke less empathy, there was also no significant difference in empathy/sympathy ratings between the onset controllable and uncontrollable conditions”. Should the authors find this proposition relevant, they might briefly comment on it in discussion.

Thank you for this insightful suggestion. In response, we have revised our Discussion section (lines 630-642) to incorporate the possibility that participants may have perceived the behaviour leading to the injury in our onset controllable condition (i.e., failing to wear protective equipment) as relatively minor and less morally reprehensible than the behaviours used in prior studies (e.g., driving while intoxicated; cocaine dependency). We acknowledge that this difference in the moral salience and perceived culpability of the antecedent behaviour may have reduced the downstream impact of the onset controllability manipulation in our study, such as sentencing and empathy-related judgements.

(3) In lines 258-259 there is a small typing error: “Care was taken was avoid providing any specific motivations...”

This has been corrected accordingly, to “Care was taken to avoid providing any specific motivations….” (lines 256-257).

---

## [Decision Letter · Decision Letter 2]

23 Nov 2025

Judging the Unseen: The Impact of Onset Controllability in Shaping Perceptions of Defendants with Traumatic Brain Injury

PONE-D-24-27473R2

Dear Dr. Williams,

We’re pleased to inform you that your manuscript has been judged scientifically suitable for publication and will be formally accepted for publication once it meets all outstanding technical requirements.

Kind regards,

Vincenzo Auriemma

Academic Editor

PLOS ONE

Additional Editor Comments (optional):

Reviewers' comments:

Reviewer's Responses to Questions

**Comments to the Author**

Reviewer #1: All comments have been addressed

Reviewer #2: All comments have been addressed

2. Is the manuscript technically sound, and do the data support the conclusions?

Reviewer #1: Yes

Reviewer #2: Yes

3. Has the statistical analysis been performed appropriately and rigorously?

Reviewer #1: Yes

Reviewer #2: Yes

4. Have the authors made all data underlying the findings in their manuscript fully available?

Reviewer #1: Yes

Reviewer #2: Yes

5. Is the manuscript presented in an intelligible fashion and written in standard English?

Reviewer #1: Yes

Reviewer #2: Yes

Reviewer #1: (No Response)

Reviewer #2: (No Response)

**Do you want your identity to be public for this peer review?** For information about this choice, including consent withdrawal, please see our Privacy Policy

Reviewer #1: No

Reviewer #2: No

---

## [Editor Report · Acceptance letter]

PONE-D-24-27473R2

PLOS ONE

Dear Dr. Williams,

I'm pleased to inform you that your manuscript has been deemed suitable for publication in PLOS ONE. Congratulations! Your manuscript is now being handed over to our production team.

Kind regards,

on behalf of

Dr. Vincenzo Auriemma

Academic Editor

PLOS ONE